# DenseMixer: Improving MoE Post-Training via Precise Router Gradient

## Abstract

The widely adopted Top-K router in Mixture-of-Experts (MoE) models is mathematically non-differentiable, making them harder to train than dense counterparts. We propose DenseMixer, a simple and effective MoE post-training technique that provides a more precise router gradient estimation by trading in some extra compute. It is universally applicable to $\text{TopK}$ routed MoE, operates in a plug-and-play manner, preserves inference-time efficiency, and remains fully compatible with popular training libraries as well as parameter-efficient methods such as LoRA. Extensive experiments demonstrate that DenseMixer consistently outperforms conventional approaches across MoE scales (7B–30B), architectures (with and without shared experts), pre-training regimes (from scratch and up-cycling), and post-training data types (instruction tuning and long chain-of-thought). Furthermore, our analysis on efficiency shows that the additional computation introduced by DenseMixer remains modest, providing substantial improvements at an acceptable cost.

## 1 Introduction

Mixture-of-Experts (MoE) has emerged as a powerful paradigm for scaling neural networks while maintaining computational efficiency (Comanici et al., 2025; Liu et al., 2024a; Yang et al., 2025). At the same time, MoE has been observed to be harder to train than dense models.

The challenge stems from the cornerstone of MoE – the sparse routing mechanism, which is typically implemented via a $\text{TopK}$ router that is mathematically non-differentiable (Shazeer et al., 2017; Liu et al., 2023). This issue blocks straightforward back-propagation and significantly complicates gradient computation, particularly for the router parameters that determine expert selection. The non-differentiability problem forces existing training methods to rely on imprecise gradient approximations, where the problematic $\text{TopK}$ operation is treated as having zero gradient during automatic differentiation, or to freeze router parameters entirely during fine-tuning (Unsloth AI, 2025).

To address the non-differentiability problem, as shown in Figure 1, we introduce DenseMixer for MoE post-training, where we trade additional compute (on inactive experts during the forward pass) for more precise router gradient estimation. Specifically, DenseMixer employs straight-through estimators (Bengio et al., 2013) to approximate the gradient for the $\text{TopK}$ operation, requiring outputs from all experts during training while maintaining sparse computation during inference.

Unlike previous solutions that depends on gradient approximations under restrictive routing assumptions (Wang et al., 2024b) or only works for small activated expert counts (e.g., K=1 or 2) (Liu et al., 2023), DenseMixer is universally applicable to modern MoE models that employ fine-grained $\text{TopK}$ routing and activate multiple experts (e.g., K=8). Our method can be applied in a plug-and-play post-training manner, preserves inference behavior, and remains fully compatible with existing training frameworks and parameter-efficient techniques such as LoRA (Hu et al., 2022).

To verify the effectiveness of DenseMixer, we conduct experiments on four representative MoE models spanning scales from 7B to 30B, architectures with or without shared experts, pre-training regimes from scratch or up-cycling, and post-training data types such as instruction or long chain-of-thought data. Across these settings, DenseMixer consistently outperforms conventional MoE training, Expert-Specialized Fine-Tuning (ESFT) (Wang et al., 2024a), and frozen-router approaches (Unsloth AI, 2025) on a variety of downstream tasks, under both full fine-tuning and LoRA adaptations.

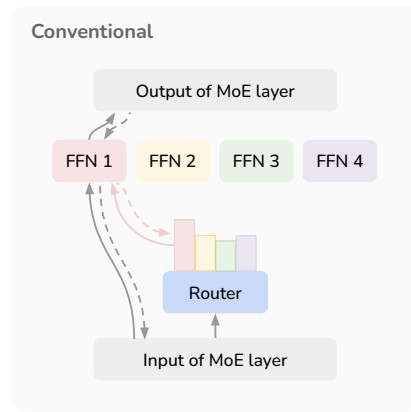 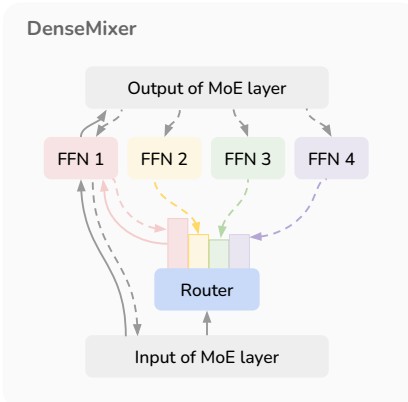

Figure 1: Comparison of router-gradient computation in MoE training. Solid arrows denote forward propagation, and dashed arrows denote backpropagation. Conventional training uses outputs from selected experts only, while DenseMixer additionally collects outputs from all experts for more accurate router gradient estimation.

We further evaluate the computational overhead introduced by DenseMixer. Although it adds approximately 46% additional FLOPs (measured on Qwen3-30B-A3B MoE), the actual time overhead is significantly lower—particularly for smaller models or moderate datasets, where the extra cost is almost negligible. This is because DenseMixer increases FLOPs only in the forward pass during post-training, while other factors such as communication often dominate runtime. In post-training scenarios, where dataset scales are much smaller than in pre-training and performance gains are prioritized over efficiency, this overhead remains well within acceptable bounds.

To summarize, our contributions are three-fold:

- We propose DenseMixer, a novel MoE post-training technique that provides a more precise router gradient estimation thus better performance through the straight-through estimator.
- We provide comprehensive empirical validation across multiple MoE scales, architectures, and training scenarios, demonstrating consistent improvements over conventional methods.
- We offer a plug-and-play implementation that requires minimal code changes and is compatible with existing training frameworks, enabling easy adoption for practitioners.

## 2 PRELIMINARIES

In this section, we provide background on MoE and its training challenges arising from the non-differentiability issue (§ 2.1), and revisit how conventional method attempts to address it, highlighting the limitations (§ 2.2) that motivate our DenseMixer approach.

### 2.1 MOE TRAINING CHALLENGES

In Transformer-based MoE models, the standard feed-forward network (FFN) layer is replaced by a Mixture-of-Experts (MoE) layer, consisting of a set of $N$ parallel FFNs referred to as experts: $\{E_0(x), E_1(x), \ldots, E_{N-1}(x)\}$. A lightweight router network – typically a linear layer parameterized by $\theta$ – followed by a softmax produces routing probabilities and dynamically selects a subset of experts to activate for each input $x$ (Jacobs et al., 1991; Jordan & Jacobs, 1994; Shazeer et al., 2017).

**MoE Forward Propagation.** The forward pass consists of the following stages.
(1) The routing weight for each expert $E_i$ is computed with the softmax function as

$$\pi_i = \text{softmax}(x \cdot \theta^T)_i. \tag{1}$$

(2) The output of the MoE layer is computed as the weighted sum of expert outputs, with $\text{TopK}$ selection activating only the highest-scoring experts:

$$y = \sum_{i=0}^{N-1} \pi_i \cdot \text{TopK}(\pi)_i \cdot \text{Expert}_i(x), \tag{2}$$

where:

$$\text{TopK}(\pi)_i = \begin{cases} 1, & \text{if } \pi_i \text{ is among the top K largest values of } \pi, \\ 0, & \text{otherwise.} \end{cases} \tag{3}$$

(3) The final loss is computed using a differentiable loss function:

$$\mathcal{L} = L(y). \tag{4}$$

The forward computation proceeds much like a standard network and scales efficiently because each token activates only a small subset of experts. However, the discrete $\text{TopK}$ routing makes the selection step non-differentiable and introduces training challenges to the following backward pass.

**MoE Backward Propagation.** The gradient computation for all components except the router follows the same procedure as dense models and can be handled by standard backpropagation. The main challenge lies in computing the router's gradient, as it involves non-differentiable operations. To backpropagate from the expert output $y$ to the router parameters $\theta$, we apply the chain rule:

$$\nabla_\theta \mathcal{L} = \frac{\partial \mathcal{L}}{\partial y} \cdot \frac{\partial y}{\partial \theta} = \sum_{j=0}^{N-1} \sum_{i=0}^{N-1} \frac{\partial \mathcal{L}}{\partial y} \cdot \text{Expert}_i(x) \cdot \frac{\partial (\pi_i \cdot \text{TopK}(\pi)_i)}{\partial \pi_j} \cdot \frac{\partial \pi_j}{\partial \theta} \tag{5}$$

**Non-Differentiable Routing Problem.** This problem occurs when calculating the router gradient. 1. In eq. (5), the last term is straightforward to compute:

$$\frac{\partial \pi_j}{\partial \theta} = \frac{\partial \, \text{softmax}(x \cdot \theta^T)_j}{\partial \theta} \tag{6}$$

2. The middle term in eq. (5) expands to:

$$\frac{\partial (\pi_i \cdot \text{TopK}(\pi)_i)}{\partial \pi_j} = \underbrace{\pi_i \cdot \frac{\partial \text{TopK}(\pi)_i}{\partial \pi_j}}_{\text{term1}} + \underbrace{\text{TopK}(\pi)_i \cdot \delta_{ij}}_{\text{term2}} \tag{7}$$

where:

$$\delta_{ij} = \begin{cases} 1, & i = j, \\ 0, & i \neq j. \end{cases} \tag{8}$$

Clearly, the $\text{TopK}$ mask is non-differentiable — specifically, the term $\frac{\partial \text{TopK}(\pi)_i}{\partial \pi_j}$ is not defined. Therefore, the router's gradient is not straightforward to compute, blocking standard gradient-based optimization for router parameters (Shazeer et al., 2017; Lepikhin et al., 2021; Fedus et al., 2022).

## 2.2 HOW CONVENTIONAL METHOD WORKS

Conventional MoE training method sidesteps the non-differentiable routing issue by treating $\text{TopK}(\pi)$ as constant during backpropagation, thus neglecting the non-differentiable term $\frac{\partial \text{TopK}(\pi)_i}{\partial \pi_j}$, namely it adopts the following approximation:

$$\frac{\partial \text{TopK}(\pi)_i}{\partial \pi_j} \approx 0 \tag{9}$$

Thus, the non-differentiable term becomes:

$$\frac{\partial (\pi_i \, \text{TopK}(\pi)_i)}{\partial \pi_j} \approx \text{TopK}(\pi)_i \cdot \delta_{ij} \tag{10}$$

And finally, the router gradient is computed as:

$$\nabla_\theta \mathcal{L} \approx \nabla_{\text{Conventional}} = \sum_{j=0}^{N-1} \sum_{i=0}^{N-1} \frac{\partial \mathcal{L}}{\partial y} \cdot \text{Expert}_i(x) \cdot \text{TopK}(\pi)_i \cdot \delta_{ij} \cdot \frac{\partial \pi_j}{\partial \theta} \tag{11}$$

$$= \sum_{j=0}^{N-1} \frac{\partial \mathcal{L}}{\partial y} \cdot \text{Expert}_j(x) \cdot \text{TopK}(\pi)_j \cdot \frac{\partial \pi_j}{\partial \theta} \tag{12}$$

Although widely used, the conventional surrogate retains only one of the two terms in the router-gradient decomposition as shown in Eq. 2, discarding the selection-sensitivity term $\pi_i \cdot \frac{\partial \text{TopK}(\pi)_i}{\partial \pi_j}$. While often empirically effective, it is inherently biased — leaving clear room for more faithful and effective estimators for such gradient computation.

## 3 DENSEMIXER: TRADE COMPUTE FOR GRADIENT PRECISION

In this section, we present DenseMixer, our solution to address the non-differentiability problem in MoE training. We first introduce how our DenseMixer approach works with the straightforward estimator (§ 3.1) and then provide an analysis of its methodology design(§ 3.2).

### 3.1 DENSEMIXER WALKTHROUGH

To bridge discrete variables and backpropagation make the gradient approximation more accurate, DenseMixer adopts the **straight-through estimator** (STE) (Bengio et al., 2013).

$$\frac{\partial \text{TopK}(\pi)_i}{\partial \pi_j} \approx \delta_{ij} \tag{13}$$

This means:

$$\frac{\partial(\pi_i \, \text{TopK}(\pi)_i)}{\partial \pi_j} \approx (\pi_i + \text{TopK}(\pi)_i) \cdot \delta_{ij} \tag{14}$$

Finally:

$$\nabla_\theta \mathcal{L} \approx \nabla_{\text{DenseMixer}} = \sum_{j=0}^{N-1} \sum_{i=0}^{N-1} \frac{\partial \mathcal{L}}{\partial y} \cdot \text{Expert}_i(x) \cdot (\pi_i + \text{TopK}(\pi)_i) \cdot \delta_{ij} \cdot \frac{\partial \pi_j}{\partial \theta} \tag{15}$$

$$= \sum_{j=0}^{N-1} \frac{\partial \mathcal{L}}{\partial y} \cdot \text{Expert}_j(x) \cdot (\pi_j + \text{TopK}(\pi)_j) \cdot \frac{\partial \pi_j}{\partial \theta} \tag{16}$$

Intuitively, STE pretends that the $\text{TopK}$ selection is an identity function with respect to the router logits during backpropagation. In practice, we implement this by: 1. Forward — computing with the original hard $\text{TopK}$ as before; 2. Backward — overriding the gradient of the $\text{TopK}$ node to be the identity.

**Handling Normalized TopK.** Some recently released MoE models (Yang et al., 2025) adopt `normalized_topk_prob` implementation, which normalizes the expert weights as follows:

$$y_{\text{normalized}} = \sum_{i=0}^{N-1} \frac{\pi_i \cdot \text{TopK}(\pi)_i}{\sum_{k=0}^{N-1} \pi_k \cdot \text{TopK}(\pi)_k} \cdot \text{Expert}_i(x) \tag{17}$$

Such normalization makes the gradient computation even more complicated as both the numerator and denominator have the non-differentiable $\text{TopK}$ term. We address this by letting gradients flow through the normalization denominator only for Top-K experts, while treating it as a constant (detached) for non-Top-K experts, providing a practical solution for modern MoE architectures.

## 3.2 DenseMixer Analysis

The core idea of DenseMixer can be summarized as trading in extra compute for more precise estimation for the router gradient. Different from conventional method that backpropagates only through the active experts, DenseMixer enables the MoE model to perform backpropagation through all experts to achieve a better approximation of the router gradient.

**Computation Overhead.**  A natural drawback of using DenseMixer's gradient approximation ($\nabla_{\text{DenseMixer}}$) is that it requires the outputs of all experts (i.e., $\text{Expert}_j(x)$ for all $j \in [0, N-1]$) for a given input $x$, while conventional method's gradient approximation ($\nabla_{\text{Conventional}}$) only requires those of the TopK selected experts. This means that DenseMixer requires the MoE layer to be densely activated during forward pass, which introduces more compute than the normal case.

Nevertheless, in the context of MoE post-training, computational cost is typically less critical than in large-scale pre-training. DenseMixer therefore provides a practical trade-off: it incurs only a single additional forward pass on those inactive experts — distinct from simply setting $K$ equal to the total number of experts — and achieves improved router gradients at a manageable cost. This makes the additional overhead acceptable in practice, considering the performance gain it brings.

**Compatibility with Existing MoEs.**  Previous methods for addressing router non-differentiability, such as SparseMixer (Liu et al., 2023) and ReMoE (Wang et al., 2024b), rely on models being pretrained with specialized routing mechanisms. This limits their applicability to post-training or fine-tuning, since most modern open-source MoEs are pre-trained with the standard TopK router. Moreover, sampling-based approaches like SparseMixer, which replace TopK with stochastic sampling, are only effective when $K$ is very small (e.g., 1–2). They fail to scale to modern MoEs that typically employ fine-grained experts with much larger activation sizes (e.g., $K = 8$, 16, or more).

In contrast, DenseMixer is directly compatible with TopK routed MoEs, supports arbitrary numbers of activated experts, and delivers consistent post-training improvements without requiring any changes to the pre-training procedure. This plug-and-play property enables us to further provide an easy-to-use implementation package that can be seamlessly applied to existing MoE models.

## 4 Experiments

In this section, we verify the effectiveness of DenseMixer across multiple MoE models and training scenarios. We first introduce the experimental setup (§ 4.1), then present comparative results (§ 4.2), and conclude with efficiency analysis (§ 4.3).

## 4.1 Experimental Setup

We evaluate DenseMixer across various settings and compare it with representative baselines.

**Models.**  To validate the generality of DenseMixer, we select four base models pretrained with diverse configurations, including OLMoE-1B-7B (Muennighoff et al., 2025a), Qwen1.5-MoE-A2.7B[1], DeepSeek-V2-Lite (DeepSeek-AI, 2024), and Qwen3-30B-A3B-Base (Yang et al., 2025). The detailed configurations of these models are summarized in Table 1.

Table 1: Information of base MoE models used in our experiments.

| Model Name | Active Param. | Total Param. | Active/Total Expert Num. | Shared Expert Num. | Context Length | Normalize TopK Prob | Training Strategy |
|---|---|---|---|---|---|---|---|
| OLMoE-1B-7B | 1B | 7B | 8 / 64 | 0 | 4k | False | from scratch |
| Qwen1.5-MoE-A2.7B | 2.7B | 14B | 8 / 64 | 4 | 8k | False | up-cycling |
| DeepSeek-V2-Lite | 2.4B | 16B | 6 / 64 | 2 | 32k | False | from scratch |
| Qwen3-30B-A3B-Base | 3B | 30B | 8 / 128 | 0 | 32k | True | from scratch |

**Datasets.**  We adopt two data configurations: a *general task* configuration for lightweight backbones and a *long CoT* configuration for the stronger backbone model.

---

[1] https://huggingface.co/Qwen/Qwen1.5-MoE-A2.7B

- **General tasks.** For OLMoE-1B–7B, Qwen1.5-MoE-A2.7B, and DeepSeek-V2-Lite, we follow the training and test sets from (Wang et al., 2024a). The training set covers GSM8K (math) (Cobbe et al., 2021), CodeAlpaca (Chaudhary, 2023), intent classification, law, summarization, and translation (Wang et al., 2024a). The test set covers GSM8K (math) (Cobbe et al., 2021), MBPP(Austin et al., 2021), and HumanEval (coding) (Chen et al., 2021), as well as intent classification, law, summarization, and translation (Wang et al., 2024a).
- **Long CoT.** For Qwen3-30B-A3B-Base, due to its improved pretraining techniques, post-training on the tasks above yields limited gains. Therefore, we focus on solving more challenging competition-level math and coding questions by generating long chain-of-thoughts (CoT). Specifically, for math reasoning, we train the model with a filtered subset of the Stanford S1 dataset (Muennighoff et al., 2025b) (about 1,000 examples with reasoning trajectories distilled from DeepSeek-R1 (DeepSeek-AI, 2025)), and evaluate the model on GPQA (Rein et al., 2024), AIME (Mathematical Association of America, 2025), MATH-500 (Hendrycks et al., 2021), OlympiadBench (He et al., 2024). For coding, we train the model with a filtered subset of the Llama-Nemotron Post-Training Dataset (Nathawani et al., 2025) (about 35,000 coding examples), and evaluate the model on HumanEval(Chen et al., 2021), HumanEval+ (Team, 2023–2025), MBPP (Austin et al., 2021), and LiveCodeBench (Jain et al., 2024).

The training and evaluation datasets statistics of the general tasks are shown below and those of the Long CoT tasks are provided in Appendix § C.

Table 2: Training and test sample sizes for different datasets.

|  | GSM | MBPP | HumanEval | Intent | Law | Summarization. | Translation. |
|---|---|---|---|---|---|---|---|
| #Training Samples | 7473 | 22000 | 22000 | 7280 | 927 | 19587 | 11639 |
| #Test Samples | 1319 | 500 | 164 | 500 | 100 | 100 | 100 |

**Baselines Training Methods.** We compare DenseMixer against several representative MoE training methods that have been widely adopted.

- **Conventional**: Standard MoE training method which treat the non-differentiable Top-K operation is treated as having zero gradient in automatic differentiation.
- **ESFT**: Expert-Specialized Fine-Tuning proposed by DeepSeekAI, which updates only a pre-selected subset of experts for a given task. We include both ESFT-gate (selection based on average gating scores) and ESFT-token (selection based on token-selection ratios).
- **Frozen Router**: Freeze the router and only update other components in MoE, as suggested by practitioners for stability.

We also compare these methods in parameter-efficient fine-tuning (PEFT) settings, where we apply LoRA to all modules except the router.

**Evaluation Setup.** For each method, we conduct a grid search on training hyperparameters (learning rate and batch size) and report the best performance. We evaluate models across multiple downstream tasks relevant to each model's domain, measuring task-specific metrics such as accuracy for classification tasks and pass rates for coding tasks. Specifically, for the long CoT task, we report the evaluation metrics under three sets of decoding hyperparameters (temperature and top_p) due to the instability of long CoT evaluation. The hyperparameters are provided in § B.

## 4.2 MAIN RESULTS

Extensive experimental results demonstrate that DenseMixer consistently outperforms conventional MoE training across MoE models of varying scales, architectures, and pre-training recipes, as well as across diverse training and evaluation datasets. This performance advantage holds for both full fine-tuning and parameter-efficient fine-tuning methods (e.g., LoRA). This is shown by Table 3 under General Tasks configuration, and Table 4 and Table 5 under Long CoT configuration.

Such performance improvement remains robust under different hyper-parameter settings for LLM generation, for example, under different popular temperature and Top-p combinations. This is demonstrated extensively by Table 4 and Table 5.

Table 3: **Post-training results on general tasks with three MoE backbones.** We highlight the best score for each task in dark green and the second in light green for each setting. DenseMixer consistently achieves the best average and most per-task scores across all models, under both full-parameter finetuning and LoRA finetuning settings.

| Backbone (Total Param.) | Method | GSM | MBPP | HE | Intent | Law | Sum. | Trans. | Avg |
|---|---|---|---|---|---|---|---|---|---|
| OLMoE (7B) | Base Model | 15.85 | 19.80 | 10.97 | 0.20 | 5.70 | 7.40 | 11.09 | 10.14 |
| | Frozen Router | 44.88 | 17.80 | 7.23 | 72.80 | 22.50 | 36.05 | 28.29 | 32.79 |
| | Conventional | 45.94 | 23.40 | 18.92 | 74.60 | 22.35 | 35.99 | 26.89 | 35.44 |
| | **DenseMixer** | 49.00 | 25.12 | 20.73 | 77.40 | 23.02 | 40.64 | 32.55 | 38.35 |
| | Frozen Router +LoRA | 45.03 | 24.20 | 17.07 | 55.80 | 21.30 | 37.70 | 28.19 | 32.76 |
| | Conventional +LoRA | 44.58 | 24.20 | 15.85 | 60.20 | 21.60 | 37.30 | 26.22 | 32.85 |
| | **DenseMixer**+LoRA | 45.38 | 26.20 | 16.48 | 66.60 | 24.70 | 40.80 | 29.43 | 35.66 |
| | ESFT-gate | 43.06 | 20.80 | 14.02 | 21.20 | 22.39 | 19.50 | 17.37 | 22.62 |
| | ESFT-token | 43.82 | 19.60 | 12.80 | 20.80 | 22.60 | 17.80 | 16.67 | 22.01 |
| Qwen1.5-MoE (14B) | Base Model | 38.69 | 38.84 | 32.31 | 16.83 | 18.20 | 28.29 | 16.53 | 27.10 |
| | Frozen Router | 53.37 | 35.20 | 37.10 | 82.20 | 33.01 | 38.29 | 32.75 | 44.56 |
| | Conventional | 53.42 | 34.60 | 36.43 | 81.80 | 29.25 | 37.80 | 33.02 | 43.76 |
| | **DenseMixer** | 55.16 | 35.40 | 39.68 | 83.40 | 33.83 | 40.56 | 33.90 | 45.99 |
| | Frozen Router +LoRA | 46.77 | 31.40 | 36.58 | 71.00 | 30.30 | 30.19 | 28.08 | 39.19 |
| | Conventional +LoRA | 43.89 | 34.00 | 38.41 | 64.80 | 28.80 | 37.99 | 26.14 | 39.15 |
| | **DenseMixer**+LoRA | 47.24 | 35.40 | 38.41 | 71.80 | 31.80 | 40.20 | 29.25 | 42.01 |
| | ESFT-gate | 50.72 | 34.00 | 36.59 | 76.40 | 27.10 | 35.89 | 28.49 | 41.31 |
| | ESFT-token | 52.76 | 35.80 | 37.20 | 76.00 | 28.20 | 33.39 | 28.86 | 41.74 |
| DeepSeek-V2-Lite (16B) | Base Model | 19.00 | 43.00 | 27.44 | 3.00 | 14.90 | 16.50 | 16.20 | 12.65 |
| | Frozen Router | 32.95 | 46.60 | 31.11 | 69.30 | 28.10 | 43.70 | 24.80 | 41.48 |
| | Conventional | 48.50 | 46.00 | 31.71 | 81.80 | 29.70 | 43.00 | 24.30 | 44.70 |
| | **DenseMixer** | 51.50 | 47.00 | 32.32 | 82.40 | 32.10 | 45.80 | 25.36 | 46.42 |
| | Frozen Router +LoRA | 48.06 | 45.20 | 34.76 | 68.40 | 24.10 | 46.20 | 32.60 | 42.83 |
| | Conventional +LoRA | 50.60 | 46.50 | 35.66 | 70.20 | 22.90 | 39.90 | 28.80 | 40.45 |
| | **DenseMixer**+LoRA | 52.16 | 46.80 | 36.59 | 71.00 | 24.30 | 47.80 | 33.80 | 44.23 |
| | ESFT-gate | 29.49 | 43.90 | 28.05 | 25.20 | 16.80 | 15.20 | 17.20 | 15.51 |
| | ESFT-token | 28.66 | 43.80 | 26.22 | 26.00 | 18.00 | 16.00 | 17.21 | 19.30 |

Here, "HE" is short for HumanEval, "Sum." is for Summary, and "Trans." represents Translation.

Interestingly, as shown in Table 3, we find that Frozen Router can occasionally outperform Conventional Training, where the router is trained end-to-end with imprecise gradients. However, it consistently underperforms DenseMixer, which benefits from more accurate gradient signals for the router. Therefore, we confirm that the routing training is important in MoE optimization.

Table 4: Results on post-training Qwen3-30B-A3B-Base across different decoding configurations (Math Benchmarks). DenseMixer consistently surpasses conventional training methods.

| Method | Temp & Top-p | GPQA (avg@8) | AIME2024 (avg@32) | AIME2025 (avg@32) | Olympiad (avg@1) | MATH500 (avg@1) |
|---|---|---|---|---|---|---|
| Base Model | 0.6 & 0.95 | 38.88 | 20.63 | 7.71 | 34.81 | 72.80 |
| | 0.7 & 0.8 | 39.89 | 20.53 | 8.33 | 33.92 | 75.40 |
| | 1.0 & 0.7 | 36.36 | 18.75 | 8.75 | 31.70 | 68.00 |
| Conventional | 0.6 & 0.95 | 54.80 | 61.56 | 45.63 | 57.33 | 93.40 |
| | 0.7 & 0.8 | 54.23 | 61.67 | 44.27 | 55.41 | 92.20 |
| | 1.0 & 0.7 | 56.55 | **63.65** | 46.15 | **59.11** | 93.00 |
| **DenseMixer** | **0.6 & 0.95** | **58.52** | **63.85** | **45.83** | **58.51** | **93.60** |
| | **0.7 & 0.8** | **55.80** | **63.13** | **45.31** | **57.18** | **93.00** |
| | **1.0 & 0.7** | **58.14** | 62.71 | **47.50** | 57.77 | **93.80** |

Table 5: Results on post-training Qwen3-30B-A3B-Base across different decoding configurations (Code Benchmarks and Average). DenseMixer consistently outperforms conventional training.

| Method | Temp & Top-p | HumanEval (avg@4) | HumanEval+ (avg@4) | (MBPP (avg@1) | LiveCodeBench (avg@4) | Avg |
|--------|--------------|-------------------|--------------------|----------------|-----------------------|-----|
| Base Model | 0.6 & 0.95 | 65.24 | 60.06 | 53.60 | 16.85 | 48.94 |
|  | 0.7 & 0.8 | 62.80 | 61.12 | 55.60 | 16.48 | 49.00 |
|  | 1.0 & 0.7 | 62.65 | 59.14 | 49.80 | 13.26 | 46.21 |
| Conventional | 0.6 & 0.95 | 92.23 | 86.89 | 80.80 | 32.26 | 67.21 |
|  | 0.7 & 0.8 | 91.01 | 85.37 | 76.80 | 29.39 | 65.59 |
|  | 1.0 & 0.7 | 90.85 | 86.59 | 79.00 | 33.42 | 67.59 |
| **DenseMixer** | **0.6 & 0.95** | **93.59** | **89.02** | **82.00** | **34.31** | **68.80** |
|  | **0.7 & 0.8** | **91.92** | **86.89** | **80.80** | **31.89** | **67.32** |
|  | **1.0 & 0.7** | **93.29** | **88.87** | **84.39** | **34.40** | **68.99** |

## 4.3 EFFICIENCY ANALYSIS

**Computational Overhead.** Though DenseMixer requires extra compututation on inactive experts, the time consumption does not grow linearly with the number of experts. It only requires an extra forward pass and does not require gradient computation and backward parameter update on inactive experts. For Qwen3-30B-A3B, DenseMixer incurs approximately 1.46× the FLOPs compared to

Table 6: FLOPs and memory usage analysis for Qwen3-30B-A3B.

| Method | Fwd TFLOPs/layer | Bwd TFLOPs/layer | Total TFLOPs/layer | Peak mem (GB) |
|--------|------------------|------------------|--------------------|---------------|
| Conventional | 16.85 | 33.70 | 50.54 | 157.93 |
| DenseMixer | 40.04 | 33.70 | 73.74 | 164.96 |
| **Ratio (DM/Conv)** | **2.38×** | **1.00×** | **1.46×** | **1.04×** |

*Note.* Fwd = Forward, Bwd = Backward.

conventional training. The detailed FLOPs analysis in table 6 shows that DenseMixer mainly requires more FLOPs in the forward pass. Overall, the extra computation is acceptable during post-training stage, and the peak memory usage does not increase substantially.

**Training Time Comparison.** We provide actual training time comparisons across different scales of model datasets.

Table 7: Training time comparison indicating acceptable overhead in post-training scenarios.

| Model | Dataset | Conventional | DenseMixer | Overhead |
|-------|---------|--------------|------------|----------|
| Qwen1.5-MoE (14B) | Intent (7K) | 22 min | 24 min | 9% |
|  | Law (1K) | 8.5 min | 9.5 min | 12% |
|  | Summary (19K) | 1.2 h | 1.4 h | 17% |
|  | Translation (11K) | 39 min | 45 min | 15% |
| Qwen3-MoE (30B) | S1 (1K) | 2.8 h | 3.6 h | 29% |
|  | Nemotron-Code (35K) | 21 h | 28 h | 33% |

As shown in 7, the training time overhead of DenseMixer is $<20\%$ on Qwen1.5-MoE, and $\sim 30\%$ on Qwen3-MoE. Though increased with the size of model and training data, it is still acceptable in most post-training scenarios.

Overall, the results demonstrate that DenseMixer consistently outperforms conventional MoE training methods across different model scales, architectures, and training scenarios, while maintaining reasonable computational overhead suitable for post-training applications.

## 5   RELATED WORK

Current approaches to MoE training handle the non-differentiability problem through several paradigms, each with inherent limitations.

SparseMixer (Liu et al., 2023) and GrinMoE (Liu et al., 2024b) use numerical ODE methods with the mid-point method to provide scalable gradient approximations for expert routing while maintaining sparse computation. However, both methods require sampling over the routing distribution, which works well when few experts are activated (1-2) but becomes computationally challenging and difficult to implement efficiently for modern MoEs (e.g., Qwen3-MoE (Yang et al., 2025)) that typically activate many more experts (8, 16, or higher). DenseMixer does not have such restriction and proves to work well on 8 active experts in the training of modern MoEs.

DS-MoE (Pan et al., 2024) employs dense computation across all experts during training combined with sparse inference, requiring dense activation during both forward and backward passes, which is much slower,while our DenseMixer only requires dense activation during the forward pass. Because the dominant training time cost lies in the backward pass, and DenseMixer does not require dense activations in this phase, it is significantly more efficient than DS-MoE. DefaultMoE (Panda et al., 2025) shares a similar philosophy by maintaining sparse training while providing dense gradients through substituting inactive expert outputs with exponential moving averages of previously computed expert values in the same batch. However, such approximations are less precise than DenseMixer's direct computation approach, and notably, the performance improvements DefaultMoE achieves after pre-training on large token volumes are comparable to or smaller than what DenseMixer achieves through post-training on significantly fewer tokens. Additionally, DefaultMoE focuses on pre-training scenarios where computational efficiency is relatively critical, while we target the more practical post-training setting where the extra computational cost is more acceptable in exchange for improved gradient precision and consistently better performance.

## 6   CONCLUSION

We confirm that the non-differentiability of Top-K routing is a key obstacle to effective MoE training. By trading an extra forward pass on inactive experts, DenseMixer enables more precise routing gradients, consistently improving post-training quality as measured by downstream performance, beyond conventional MoE approaches under various settings.

Our comprehensive evaluation across four MoE models (7B, 14B, 16B, 30B) and diverse architectures demonstrates DenseMixer's effectiveness and generalizability. The method delivers consistent gains across pre-training regimes (from scratch vs. up-cycling), downstream tasks (math, coding, language understanding), and both full fine-tuning and parameter-efficient settings. Moreover, the added compute is modest and limited to the post-training stage, making the cost acceptable. With its plug-and-play design and compatibility with existing frameworks, DenseMixer is simple to adopt and broadly applicable for advancing MoE post-training.

## ETHICS STATEMENT

All datasets used in this work (GSM8K (Cobbe et al., 2021),MBPP (Austin et al., 2021),HumanEval (Chen et al., 2021), Intent classification, Law, Summary, and Translation (Wang et al., 2024a), Stanford S1 (Muennighoff et al., 2025b), Llama-Nemotron-Post-Training-Dataset (Nathawani et al., 2025) ) are publicly available academic benchmarks that do not contain personally identifiable or sensitive information. Our study focuses on post-training mixture-of-experts (MoE) models and does not involve the collection of new human-subject data. As with other LLM fine-tuning methods, models trained with DenseMixer may still produce incorrect, biased, or misleading content, and code-related outputs may contain insecure or faulty patterns. Our method does not eliminate these risks, and deployments in high-stakes settings should include appropriate safeguards (e.g., content filtering, human oversight, and domain-specific evaluations). The potential societal benefits of this work include improved performance of MoE models, which may help humans in various domains. This research was conducted in accordance with the ICLR Code of Ethics, and the authors take full responsibility for the analyses and conclusions presented in this paper.

REPRODUCIBILITY STATEMENT

We have taken several steps to ensure the reproducibility of our results. We applied DenseMixer on representative MoE models spanning 7B–30B parameters (Yang et al., 2025; DeepSeek-AI, 2024; Muennighoff et al., 2025a) and across diverse settings (with/without shared experts, different pre-training regimes, and multiple downstream tasks, including GSM8K (Cobbe et al., 2021),MBPP (Austin et al., 2021)),HumanEval (Chen et al., 2021), Intent classification, Law, Summary, and Translation (Wang et al., 2024a), MATH-500 (Hendrycks et al., 2021),AIME 2024 and AIME 2025 (Mathematical Association of America, 2025), consistently improving over conventional MoE training, Expert-Specialized Fine-Tuning (Wang et al., 2024a), and frozen-router approaches(Unsloth AI, 2025) under both full fine-tuning and parameter-efficient adaptationHu et al. (2022). Because the long chain-of-thought reasoning model is sensitive to decoding hyperparameters, we report results under multiple combinations of temperature and top-$p$ to demonstrate the robustness and reproducibility of our method. We detail the method specification, and we provide the practical recipe used in our experiments. An efficiency analysis reports the additional compute introduced during post-training, with a forward-only overhead and a measured FLOP ratio that remains acceptable for post-training use. Together, these descriptions enable reproduce using standard training libraries. With the provided code and instructions, our results can be reproduced using 8×H200 GPUs or equivalent hardware.

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

## A    USE OF LLMS DISCLOSURE

We disclose the following uses of large language models in the preparation of this work. GPT-5 (OpenAI, 2025) was used exclusively for language refinement and improving the manuscript's readability. In addition, Claude Code (Anthropic, 2025) served as a coding assistant to generate and debug experimental scripts. At no time did LLMs contribute to the core research ideas, methodology, or the interpretation of results. All scientific contributions, analyses, and conclusions are the responsibility of the authors. Outputs produced by LLMs were carefully reviewed and edited where necessary to ensure accuracy and integrity.

## B    HYPERPARAMETERS

We use the following hyperparameters across different models. Experiments were run on $2\times$A100, $4\times$H200, or $8\times$H200 GPUs (depending on the setting). We set `warmup_ratio` = 0.1 and use bf16 as the data type. Using these settings, our results can be reproduced on the same or equivalent hardware. Learning rate, batch size and training epochs are shown as follows:

## B.1 DEEPSEEK-V2-LITE

### B.1.1 FULL FINE-TUNING

- Learning rate: $1 \times 10^{-6}$ to $3 \times 10^{-5}$
- Epochs: 3-4 epochs
  - Most NLP tasks: 3 epochs
  - Mathematical and code tasks: 4 epochs
- Batch size: 128

### B.1.2 LORA FINE-TUNING

- Learning rate: $2 \times 10^{-4}$ to $6 \times 10^{-4}$
  - Most tasks: $2 \times 10^{-4}$
  - Translation tasks: $5 \times 10^{-4}$
  - Intent classification: $6 \times 10^{-4}$
- Epochs: 3-4 epochs
  - Most tasks: 3 epochs
  - GSM and code tasks: 4 epochs
- Batch size: 64

## B.2 QWEN1.5 MOE

### B.2.1 FULL FINE-TUNING

- Learning rate: $1 \times 10^{-6}$ to $5 \times 10^{-5}$
  - Lower range: $1 \times 10^{-6}$ to $5 \times 10^{-6}$ (most common)
  - Mid range: $1 \times 10^{-5}$ to $2 \times 10^{-5}$
  - High range: $3 \times 10^{-5}$ to $5 \times 10^{-5}$ (rare, GSM tasks)
- Epochs: 3-4 epochs
  - Summary and translation tasks: 3 epochs
  - Most other tasks: 4 epochs
- Batch size: 64

### B.2.2 LORA FINE-TUNING

- Learning rate: $5 \times 10^{-6}$ to $5 \times 10^{-4}$
  - Most tasks: $2 \times 10^{-4}$ to $3 \times 10^{-4}$
- Epochs: 3-4 epochs
  - Most tasks: 3 epochs
  - Code and GSM tasks: 4 epochs
- Batch size: 64

## B.3 QWEN3-30B

### B.3.1 FULL FINE-TUNING

- Learning rate: $1 \times 10^{-5}$ to $3 \times 10^{-5}$
- Epochs: 5 epochs
- Batch size: 4 per device with gradient accumulation of 2 (effective batch size: 8)

## B.4 OLMoE

### B.4.1 FULL FINE-TUNING

- Learning rate: $1 \times 10^{-6}$ to $2 \times 10^{-5}$
  - Code tasks: $1 \times 10^{-6}$
  - Most tasks: $2 \times 10^{-5}$
- Epochs: 4 epochs
- Batch size: 128

### B.4.2 LoRA FINE-TUNING

- Learning rate: $1 \times 10^{-4}$ to $4 \times 10^{-4}$
  - Code, GSM, and summary tasks: $1 \times 10^{-4}$
  - Translation, intent, and law tasks: $4 \times 10^{-4}$
- Epochs: 4 epochs
- Batch size: 256

## C DATA STATISTICS FOR LONG CoT TASKS

The training and test sample sizes of Long Cot Tasks are shown in table 8.

Table 8: Training and test sample sizes for different datasets.

|  | AIME 2024 | AIME 2025 | MATH-500 | Olympiad Bench | GPQA_Diamond | MBPP | HumanEval | HumanEvalPlus | LiveCodeBench |
|---|---|---|---|---|---|---|---|---|---|
| #Training Samples | 1000 | 1000 | 1000 | 1000 | 1000 | 35000 | 35000 | 35000 | 35000 |
| #Test Samples | 30 | 30 | 500 | 674 | 100 | 500 | 164 | 164 | $\sim 500$ |

