# OpenReview forum: "DenseMixer: Improving MoE Post-Training with Precise Router Gradient"
_ICLR.cc/2026/Conference — Submitted to ICLR 2026_

### Official Review · Reviewer_4vjf · 2025-10-22

**Soundness:** 4
**Presentation:** 4
**Contribution:** 2
**Rating:** 6
**Confidence:** 4

**Summary:**

This paper is introducing a post-training method for MoE models to improve the estimation of the gradient of the router which conventionally uses a non-differentiable top-k operation for sparsity in the MoE layer. The core idea of the paper is simply to introduce the STE technique for the gradient of the router logits w.r.t to the top-k selection function. This requires all experts to be activated (no sparsity). As a result, all experts are used and gradients for all router logits are more precise. Furthermore, they also provide a methodology to apply the STE for renormalized MoE models (where the router scores are normalized again after the top-k procedure).

**Strengths:**

I believe the main strength of the paper is that it provides a practical solution to making pretrained MoE models more performant as this is a post-training technique. Their results on several MoE models (OlMoe, QWen, DeepSeek) definitely show improvement over benchmark tasks (even though for many tasks the improvement is quite marginal). Therefore, I believe that as far as pushing the performance of existing models goes, this is certainly a useful technique that I believe could be incorporated into standard post-training pipelines for sparse MoE models. The paper was written clearly, I did not find any mathematical or typographical errors and the efficiency / overhead analysis explains the tradeoffs in performance vs cost quite well.

**Weaknesses:**

While I certainly believe the paper will be a useful method for pushing up the performance of MoE models, my main issue is with novelty in this area. The issue of incorrect gradients in MoE models for the router has been explored a lot in the literature. As noted in the related work section and other papers, many novel techniques have been developed to get a more precise gradient. Compared to those papers, the improved estimation technique is really basic as it is a simple use of the well known STE estimator technique. I believe the only novelty in this paper is that they discovered it is good enough to only do this in post-training. If not for post-training, this technique would be very prohibitive and not very likely to be used. Regardless, the STE is the motivation for most of the 'more accurate router gradient paper' as the goal is to achieve that performance without the cost of activating all experts.

**Questions:**

The paper was written quite well and therefore I don’t really have any serious confusions or questions. My only question is how well this technique would work if incorporated during the pre-training stage as compared to in the post-training stage. Of course it would be much more expensive but I wonder if the authors did any experiments regarding this. I do not expect them to do such an experiment but would want to know if they did or what their thoughts are.

---

> ### Author Response · Authors · 2025-11-21
> **Author Response to Reviewer**
>
> We appreciate Reviewer 4vjf's deep reading and positive assessment of our paper. We address the remaining concerns and answer the questions below.
>
> **Response to Weakness**
>
> **1. Simplicity ≠ triviality — making STE work for modern MoEs is non-trivial.**
>
> While STE is an existing method, it does **not** work out-of-the-box for Top-K MoEs—especially **normalized Top-K** (e.g., Qwen3-MoE), where numerator and denominator depend on non-differentiable masks. DenseMixer introduces a targeted gradient-flow rule to make it work.
>
> We also note that ReMoE, which also based on a relatively simple mechanism (ReLU routing), was accepted to ICLR 2025, which suggests that practical and effective techniques can constitute meaningful contributions when they address real and widespread problems.
>
> **2. Prior methods cannot be applied in post-training.**
>
> ReMoE, SparseMixer, and GrinMoE require **custom routing during pretraining**, making them incompatible with the vast majority of existing Top-K MoEs (Qwen,DeepSeek,GPT). DenseMixer is **fully plug-and-play** and works directly on standard checkpoints.
>
> **3. DenseMixer provides the first broad evidence that better router gradients help post-training.**
>
> Across **7B–30B models**, different MoE architectures, multiple datasets, and both full and LoRA fine-tuning, DenseMixer delivers consistent improvements—something not shown by prior work focused on router-gradient quality.
>
> **4. Practical, empirical methods are crucial for MoE training.**
>
> MoE training remains poorly understood in the open-source community. Our experiments show that **more precise router-gradient estimation reliably improves post-training**, even for models pretrained with standard Top-K routing. These results offer a concrete empirical foundation for future efficiency-focused improvements.
>
> **5. DenseMixer is specifically designed for post-training constraints.**
>
> Post-training must preserve pretrained behavior, maintain sparse inference, and stay within smaller compute budgets. DenseMixer meets all these requirements and  directly usable for existing models.
>
> **Response to Question**
>
> We did not run pre-training experiments with DenseMixer, mainly because the computational cost of large-scale MoE pre-training is prohibitive in our setting. That said, we expect our technique to be at least as beneficial, and likely **more** beneficial, when used during pre-training: in general, having a consistent gradient propagation scheme for both pre-training and post-training should lead to better-optimized routers and experts.
>
> In other words, if DenseMixer were used throughout pre-training and post-training (rather than only in post-training), the model could learn under this richer router gradient from the start, and we anticipate this would further improve performance beyond what we observe in the post-training-only setting. We view this as an exciting direction for future work.
>
> To extend DenseMixer to full pre-training, the key challenge is **reducing its computational cost**. Conventional training computes router gradients using only **K** active experts, whereas DenseMixer activates all **N** experts. An intermediate approach—activating **more than K but fewer than N** experts—may offer a promising trade-off: richer router-gradient signals with significantly lower overhead. Exploring such partial-dense strategies is a natural next step and could make DenseMixer practical for large-scale MoE pre-training.

---

### Official Review · Reviewer_VXEj · 2025-10-31

**Soundness:** 3
**Presentation:** 3
**Contribution:** 3
**Rating:** 8
**Confidence:** 4

**Summary:**

The authors address the non-differentiability of Top-K routing in Mixture-of-Experts models. The paper proposes DenseMixer, using a straight-through estimator. DenseMixer computes hard TopK selections in the forward pass, then overrides the TopK backward pass so that the gradients flow as if the TopK was replaced with the identity function. This enables gradients to propagate through all expert outputs. The paper shows consistent empirical gains across several MoE scales, datasets, and fine-tuning methods (full and LoRA), and the improvements justify the FLOP overhead and run-time costs.

**Strengths:**

The method is practical and simple to implement, and it preserves model behavior at inference.

The empirical evidence is very strong. The DenseMixer approach outperforms standard MoE across multiple model scales and datasets.

The authors provide an efficiency analysis that justifies the increase in computational overhead relative to performance improvements.

The method is compatible with other fine-tuning tricks such as normalized TopK and LoRA for parameter efficient training.

**Weaknesses:**

The experiments lack various settings of different K and N to demonstrate how the method is affected by sparsity.

It would be convincing to see how the memory and computational overhead scales as a function of model size, e.g. to confirm that the overhead is not increasing as the model becomes larger.

**Questions:**

Can you provide a small-scale experiment where you compute a numerical finite-difference estimate (or autograd) of the router gradient (or an alternative high-fidelity gradient proxy) and compare the bias/variance of standard MoE vs. STE/DenseMixer vs. dense model? This would make the claim about improved gradient concrete.

How does the performance of DenseMixer depend on the sparsity (K)?

How sensitive are the gains to the amount of post-training data e.g. do benefits saturate quickly?

Are there any training stability issues when enabling a dense forward pass, provided the model is pretrained with sparse forward passes and the new activations are out of distribution?

---

> ### Author Response · Authors · 2025-11-21
> **Author Response to Reviewer VXEj**
>
> We thank Reviewer VXEj for their thorough review and insightful comments. We are pleased that the reviewer found DenseMixer to be a practical and simple method that preserves model behavior at inference while achieving strong empirical gains.
>
> We now address the reviewer’s concerns regarding the identified weaknesses in our experiments and analysis.
>
> **Response to Weakness 1**
>
> Table 1 (L260–266, page 5) shows the models we use have different sparsity configurations, i.e., varying combinations of the number of experts \(N\) and activated experts \(K\). To further clarify this, in the general rebuttal we additionally include experiments on a MiniCPM-MoE model with a smaller \(K\).
>
> Across all these configurations, DenseMixer consistently improves or matches the performance of standard MoE baselines, indicating that our method is robust to different choices of \(K\) and \(N\) and is not adversely affected by model sparsity.
>
> **Response to Weakness 2**
>
> We augment Table 7 with results from OLMoE. These results show that the relative overhead of DenseMixer remains in a similar range across 7B and 14B models. Instead, the overhead correlates more with the sparsity configuration (i.e., the ratio \(K/N\)) than with the total parameter count. This supports our claim that DenseMixer does not introduce disproportionately higher computation or memory costs as the model becomes larger.
>
> |Model|Dataset|Conventional|DenseMixer|Overhead|
> |---|---|---|---|---|
> |OLMoE-7B|GSM8K(7K)|6.2h|6.82h|10%|
> | |CodeAlpaca(22K)|15h|15.6h|4%|
> |Qwen1.5-14B|Intent(7K)|22min|24min|9%|
> | |Law(1K)|8.5min|9.5min|12%|
> | |Sum.(19K)|1.2h|1.4h|17%|
> | |Trans.(11K)|39min|45min|15%|
> |Qwen3-30B|S1(1K)|2.8h|3.6h|29%|
> | |NemoCode(35K)|21h|28h|33%|
>
>
> **Response to Q1**
>
> We agree that making the claim about improved router gradients more concrete is important. However, because the Top-K routing operation is non-differentiable, there is no natural “ground-truth” gradient to compare against: both autograd through the discrete Top-K and finite-difference estimates around the switching points do not provide a reliable reference for measuring bias.
>
> Instead, we directly compare the router gradients produced by conventional MoE fine-tuning and by DenseMixer. We measure (i) the mean L2 norm of the router gradients and (ii) their coefficient of variation (CV = std / mean) over tokens and steps. The results are summarized in the table:
>
> |Method|MeanL2|CV|Mean L2 diff vs. vanilla|
> |---|---|---|---|
> |Vanilla|31.33|158.17|–|
> |DenseMixer|44.59|24.33|31.65|
>
> These numbers show that DenseMixer yields router gradients with substantially larger magnitude (reflecting the contribution of non-activated experts that would otherwise receive zero gradient) and much lower relative variance. This supports our claim that DenseMixer provides stronger and more stable router gradients than conventional MoE post-training.
>
> **Response to Q2**
>
> Table 3 in the main paper already reports results on several MoE models with different sparsity. In addition, in the general rebuttal we include further experiments on MiniCPM-MoE. Across all these settings, DenseMixer consistently achieves improvements over the corresponding MoE baselines, and we do not observe a clear monotonic trend of performance with respect to \(K\). This suggests that the effectiveness of DenseMixer does not strongly depend on the exact sparsity level of the underlying MoE model.
>
> **Response to Q3**
>
> We ran additional experiments on Qwen1.5-MoE with a summarization task, varying the fraction of post-training data used while keeping the number of epochs fixed to 1. The results are shown below:
>
> |Method|0%|25%|50%|75%|100%|
> |---|---:|---:|---:|---:|---:|
> |Conventional|26.1|27.8|29.4|32.0|33.6|
> |DenseMixer|26.1|28.4|32.4|34.5|36.7|
> |Gain|0.0|+0.6|+3.0|+2.5|+3.1|
>
> We see that the improvement of DenseMixer over conventional does **not** quickly saturate at small data sizes. Instead, the gains remain substantial and generally increase as more post-training data is used (up to +3.1 points at 100%). This suggests that DenseMixer continues to benefit from additional data, rather than only providing a small boost in the very low-data regime.
>
> **Response to Q4**
>
> We have not observed training stability issues when enabling the dense forward pass. Conceptually, both the sparse (Top-K) and dense variants are approximations to the same underlying router objective, rather than two completely different distributions. As shown in our Response to Q1, DenseMixer yields router gradients with **lower variance** than conventional MoE fine-tuning, which reduces the risk of “out-of-distribution” activations causing unstable updates. Empirically, we did not see divergence or abnormally large losses when switching to DenseMixer, even though its gradient propagation differs from the original sparse pretraining.

---

### Official Review · Reviewer_7J9M · 2025-10-31

**Soundness:** 2
**Presentation:** 3
**Contribution:** 1
**Rating:** 2
**Confidence:** 4

**Summary:**

This paper addresses the challenge of post-training MoE models, where the non-differentiable nature of the top-k routing mechanism hinders effective gradient-based optimization of the router. The authors propose *DenseMixer*, a technique that applies the well-established Straight-Through Estimator (STE) to create a more precise, dense gradient signal for the router parameters during the backward pass. This is achieved by performing an additional forward pass on inactive experts during training, trading a modest increase in computation for improved performance. The method is evaluated on a diverse set of modern MoE models and consistently outperforms standard post-training baselines like conventional training  and freezing the router, all while maintaining inference-time efficiency

**Strengths:**

- The paper is clearly written and easy to follow. The main contributions are clear and backed up with experimental results.
- The authors have considered relevant datasets and benchmarks, the proposed method is well evaluated. Also, the post-training setup is relevant to the practitioners.
- The paper explicitly mentions the computational overhead introduced by DenseMixer and provides detailed wall-clock time measurements.

**Weaknesses:**

**Limited Novelty:** The primary concern is that the core technical contribution is the application of the Straight-Through Estimator (STE), a well-known technique, to MoE routers. While the authors effectively demonstrate its utility for MoE post-training, the paper does not introduce a new fundamental algorithm. The authors should reframe their contribution to be more precise. Simply showing post training would benefit from the well-established straight-through router is a not a sufficient contribution.

**Insufficient Comparison to Other Differentiable Routing Methods:** The paper’s experimental baselines are limited to simple heuristics (freezing the router) or the standard training method. It fails to compare against other sophisticated methods that have been explicitly designed to solve the same non-differentiable routing problem. The related work section mentions methods like SparseMixer and ReMoE, but these are not included as experimental baselines. The authors argue that these methods are not suitable for the post-training context or for large *K* values, but this claim is not empirically substantiated. A direct comparison is necessary to properly situate DenseMixer in the literature and validate its superiority

**Hyperparameter Sensitivity and Tuning Overhead:** The paper states that learning rate and batch size were selected via grid search, and Appendix B only provides wide ranges for these hyperparameters. This undermines the "plug-and-play" claim, as it implies that significant tuning may be required to achieve the reported results, introducing a hidden computational cost. To substantiate the method's robustness, the authors should include a *sensitivity analysis* showing how performance varies with changes in key hyperparameters.

**Justification of Cost-Benefit Trade-off:** While the overhead is well-documented, the performance gains, although consistent, are sometimes modest. For example, on several benchmarks, DenseMixer provides a 2-4% absolute improvement over conventional training in exchange for a 15-30% increase in training time

**Questions:**

- A few more implementation details could have been given. For example, what is the number of tokens used for answer generation? What exactly avg@N means in Table 4? I assume the accuracies were averaged over N runs but it would be nice to have it clearly mentioned in the text. Similarly, in Appendix B, some of hyperparameter values are given in a range (e.g., learning rate from 1x10^-6 to 3x10^-5). While these are minor details, it is very helpful to know all the hyperparameters.

- From the paragraph "Evaluation Setup" at the end of Section 3, it follows that batch size and learning rate were chosen based on the performance of the method. How robust is the model's improvement under changing LR and BS values?

---

> ### Author Response · Authors · 2025-11-21
> **Author Response to Reviewer 7J9M**
>
> We thank Reviewer 7J9M for recognizing our clear writing and comprehensive experiments.
>
> Please check out our `General Rebuttal` (**a separate reply above**) first. We address the remaining concerns as below.
>
> ---
>
> **Response to Weakness 1**
>
> **1. Simplicity ≠ triviality — making STE work for modern MoEs is non-trivial**
>
> While STE is well-known, it does **not** work out-of-the-box for Top-K MoEs—especially **normalized Top-K** (e.g., Qwen3-MoE), where both numerator and denominator have non-differentiable TopK. DenseMixer introduces a targeted gradient-flow rule to make it work.
>
> Importantly, ReMoE cited by the reviewer—also based on a simple mechanism (ReLU routing)—was accepted to ICLR 2025, showing that practical, effective techniques remain valuable contributions when they solve real problems.
>
> **2. Prior methods cannot be applied in post-training**
>
> ReMoE, SparseMixer, and GrinMoE require **custom routing during pretraining**, making them incompatible with the majority of existing Top-K MoEs (Qwen,DeepSeek,GPT). DenseMixer is **fully plug-and-play** and works directly on standard checkpoints.
>
> **3. DenseMixer provides the first broad evidence that better router gradients help post-training**
>
> Across **7B–30B models**, different MoE architectures, multiple datasets, and both full and LoRA fine-tuning, DenseMixer delivers consistent improvements—something not shown by prior work focused on router-gradient quality.
>
> **4. Practical, empirical methods are crucial for MoE training**
>
> MoE training remains poorly understood in the open-source community. Our experiments show that **more precise router-gradient estimation reliably improves post-training**, even for models pretrained with standard Top-K routing. These results offer a concrete empirical foundation for future efficiency-focused improvements.
>
> **5. DenseMixer is specifically designed for post-training constraints**
>
> Post-training must preserve pretrained behavior, maintain sparse inference, and stay within smaller compute budgets. DenseMixer meets all these requirements and  directly usable for existing models.
>
> ---
>
> **Response to Weakness 2**
>
> Please check our `General Rebuttal` above for more details.
>
> ---
>
> **Response to Weakness 3**
>
> We clarify that our goal was **not** to tune DenseMixer extensively, but rather to **maximize the baselines**. We performed a broad grid search **on the baseline** to ensure it was as strong as possible, and then directly reused those *baseline-optimal* hyperparameters for DenseMixer.
>
> DenseMixer achieved consistent gains **without any additional tuning**, indicating that its improvements do not rely on hyperparameter sensitivity. In other words, DenseMixer outperforms the *best-tuned baseline*, not a weaker or under-tuned version.
>
> In practice, using DenseMixer requires **no extra tuning cost** beyond what practitioners already perform for conventional MoE training. The hyperparameter search you would do for the baseline is exactly what you would do for DenseMixer as well. Our goal is to demonstrate that **DenseMixer’s achievable performance ceiling is higher** than that of conventional training.
>
> It is neither practical nor meaningful to require one method to outperform another under *all* hyperparameter settings, since suboptimal learning rates or batch sizes can undertrain any model. Therefore, comparing both methods at their **respective best-performing settings** is the standard and reasonable criterion—and under this criterion, DenseMixer consistently shows superior performance.
>
> ---
>
> **Response to Weakness 4**
>
> In post-train, data scale is far smaller than pretrain, and **performance gains** is prioritized over small increases in training time. DenseMixer introduces only a modest overhead. E.g., when fine-tuning **Qwen1.5-MoE (14B)**, DenseMixer adds only a few minutes per run—well within the acceptable range, especially given the performance gain.
>
> |Dataset|Conv.|DenseMixer|DefaultMoE|
> |---|---|---|---|
> |Intent (7K)|22min|24min|22.7min|
> |Law (1K)|8.5min|9.5min|8.9min|
> |Summary (19K)|1.2h|1.4h|1.28h|
> |Translation (11K)|39min|45min|42min|
>
> Overall, though DenseMixer trades some efficiency, the **absolute overhead remains small** in practical post-training and is outweighed by the **consistent performance gains**.
>
> ---
> **Response to Question 1**
>
> We will open-source all training and evaluation code, including runnable scripts to fully reproduce our results.
>
> For the max generation length, we used 20K tokens for Qwen3-MoE and 4K tokens for all other models.
>
> “AVG@N” indicates that metric is averaged over N runs to mitigate randomness. Evaluating Qwen3-MoE with a 20K token budget is particularly expensive—e.g., a single benchmark run on 8×H200 GPUs can take over one hour per checkpoint.
> Despite this cost, we carried out all repeated runs to ensure stable & reliable evaluation, and we will release the exact scripts and configurations used.
>
> **Response to Question 2**
>
> Please see response to Weakness 3 above.

---

### Official Review · Reviewer_WUwb · 2025-11-05

**Soundness:** 2
**Presentation:** 4
**Contribution:** 3
**Rating:** 4
**Confidence:** 4

**Summary:**

This paper proposes a method to improve the training of mixture-of-experts (MoE) language models by leveraging a dense forward pass to obtain the outputs of each expert while keeping the backward pass sparse. The key idea is to trade off computation during the “forward pass” for a dense router gradient during the backward pass. In their experiments, the authors evaluate their method against select baselines for supervised fine-tuning. They find that their approach yields consistent performance improvement across a range of benchmarks compared to the baselines they selected. They also measure computational and memory overhead, which is appreciated.

**Strengths:**

- Practical Technique: The method is straightforward and is therefore much more likely to be adopted by practitioners and used for large-scale training.

- Reasonable Model Scales and tasks: They tested on MoEs that are large enough to be realistic for practical use on practical datasets and benchmarks, so the results are relevant to real-world models.

- Clear Writing: The paper is well-written and the authors explain their method and results clearly, making it easy to follow.

- Hyperparameter search: The authors claim to have conducted a hyperparameter grid search on lines 308-309. This reassures me that the paper is likely to report the best possible performance of tested methods.

- Reasonable evaluation of memory and computational overhead: In tables 6 and 7, the authors provide a reasonable evaluation of the computational and memory overhead of their method.

**Weaknesses:**

- My main concern is that the authors do not compare to a clearly relevant and published baseline from the literature (e.g., DefaultMoE). As the authors mention on line 448

> DefaultMoE [1] shares a similar philosophy by maintaining sparse training while providing dense gradients through substituting inactive expert outputs with exponential moving averages of previously computed expert values in the same batch.

The authors seem to provide two reasons for not comparing (Lines 450-456):

    1. Equating the performance gains of DefaultMoE during pre-training with the performance gains of DenseMixer during post-training, although the two are not directly comparable.

    2. Stating that DefaultMoE focuses on pre-training.

I disagree with the authors as their claim (1) is made between quantities that cannot be compared: there are too many confounding factors between pre- and post-training, which could affect the claimed comparison. I also disagree that (2) is a good reason not to compare the methods since DefaultMoE can be trivially applied to post-training and will scale better in FLOP-overhead as the number of experts E is increased than DenseMixer. **Ideally, the authors should include a wall-clock and per-step comparison to DefaulMoE.**  Other relevant baselines addressing the same problem that are not compared to include [2,3], but I believe that the most relevant is [1].


- Missing limitation: poor scaling with the number of experts (E). The proposed method scales linearly in FLOP-overhead relative to conventional or published techniques from the literature that tackle the same problem (DefaultMoE [1,2]) as the number of experts (E) is increased. This is essential for post-training the largest models (e.g., DeepSeekV3 256 total experts and KimiK2 384 total experts). Ideally, the scaling should be reported in the evaluation of computational overhead.

- Misleading claim in the abstract. The following sentence makes me believe the authors evaluate their technique in a pre-training setting while this is not the case:

>Extensive experiments demonstrate that DenseMixer consistently outperforms conventional approaches across MoE scales (7B–30B), architectures (with and without shared experts), pre-training regimes (from scratch and up-cycling), and post-training data types (instruction tuning and long chain-of-thought).

I believe it would be clearer if the authors modified this to indicate that Dense Mixer can fine-tune models pre-trained in these ways.

- Although the paper states that it performed a grid search over hyperparameters, this could be trivial if the authors were to select a grid of two values. Could the authors confirm that the hyperparameters selected were interior points of the set of values tried?


[1][Dense Backpropagation Improves Training for Sparse Mixture-of-Experts; NeurIPS 2025]

[2][Grinmoe: Gradient-informed mixture-of-experts.]

[3][Dense training, sparse inference: Rethinking training of mixture-of-experts
language models.]


**Adding a comparison to defaultMoE, confirming the validity of your hyperparameter search, and adding the overhead scaling of the method would cause me to raise my score.**

**Questions:**

**Typos found:**
- Line 179 straightforward → straight-through

---

> ### Author Response · Authors · 2025-11-21
> **Author Response to Reviewer WUwb**
>
> We thank Reviewer WUwb for their thorough review and insightful comments. We are pleased that the reviewer recognized our method as a practical technique. We address the concerns below:
>
> **Response to Weakness 1**
>
> > “*My main concern is that the authors do not compare to a clearly relevant and published baseline …*”
> >
>
> To address this, we have added comparisons with **DefaultMoE**, **ReMoE**, and **SparseMixer**. Given time limit, we experiment with **Qwen1.5-MoE (14B)** and **DeepSeek-v2-Lite (16B)** backbones, and also validate the SparseMixer experiment using **MiniCPM-MoE-8x2B** (Top2 activated). We performed the same grid search for each baseline to ensure fair comparison.
>
> Please check our `General Rebuttal` above (**a separate comment reply above**) for more details. The wall-clock time comparison is also discussed there.
>
> **Response to Weakness 2**
>
> > *“… scales linearly in FLOP-overhead … as the number of experts (E) is increased …”*
> >
>
> While the FFN forward cost scales linearly with $E$, the **total** training overhead is sub-linear.
> The total training FLOPs can be decomposed as:
> $Total FLOPs=Forward_{Attn} + `Forward_{FFN}`+ Backward$
>
> - In standard training, The FLOPs of Backward $\approx$ 2 \times Forward. DenseMixer only increases `${Forward}_{FFN}$` linearly with the number of experts, while leaving the Backward pass and ${Forward}_{Attn}$ unchanged (sparse). So the **total training FLOPs** does not grow linearly with expert number.
> - Given smaller post-training datasets, the absolute overhead is marginal. For instance, on Qwen1.5-MoE (14B), training time increased by only ~2 minutes (22 $\to$ 24 min), representing a highly favorable trade-off for the consistent performance gains.
>
> **Response to Weakness 3**
>
> > *“… I believe it would be clearer if the authors modified this to indicate that Dense Mixer can fine-tune models pre-trained in these ways.”*
> >
>
> Thank you for pointing it out. We do mean finetuning models instead of pretraining, and will modify it accordingly.
>
> **Response to Weakness 4**
>
> > “*… Could the authors confirm that the hyperparameters selected were interior points of the set of values tried?*”
> >
>
> We confirm that the selected hyperparameters are indeed **interior points** within our search space. The search ranges and selected optimal values (which fall inside the ranges) are detailed in our `General Rebuttal` above (**a separate comment reply above**)
>
> We sincerely thank the reviewer for these thoughtful suggestions and hope our response can address the remaining concerns.

---

> > ### Comment · Reviewer_WUwb · 2025-11-27
> >
> > The authors have addressed my concerns and I am raising my score accordingly. I now recommend accepting the paper.

---

### Author Response · Authors · 2025-11-21
**General Rebuttal by the Authors**

We thank all reviewers for their constructive feedback. Several comments across reviewers concern (1) **comparisons with additional baselines** and (2) **hyperparameter search rigor**. We address these common questions here.

---

**1. Additional Baseline Comparisons**

Multiple reviewers suggested baselines like **DefaultMoE**, **SparseMixer**, and **ReMoE**. Given time limit, we compare them using Qwen1.5-MoE (14B) and DeepSeek-v2-Lite (16B).

- **`Results on Qwen1.5-MoE`**
    |Method|GSM|MBPP|HumanEval|Intent|Law|Sum.|Trans.|avg|
|---|---|---|---|---|---|---|---|---|
|Base|38.69|38.84|32.31|16.83|18.20|28.29|16.53|27.10|
|Frozen|53.37|35.20|37.10|82.20|33.01|38.29|32.75|44.56|
|Conv.|53.42|34.60|36.43|81.80|29.25|37.80|33.02|43.76|
|**DenseMixer**|**55.16**|**35.40**|**39.68**|**83.40**|**33.83**|**40.56**|**33.90**|**45.99**|
|DefaultMoE*|51.00|33.20|35.33|80.60|31.20|38.40|24.00|41.96|
|SparseMixer*|1.30|0.00|3.90|3.80|3.40|2.10|3.50|2.57|
|ReMoE*|46.30|33.00|36.80|60.80|25.50|25.80|16.99|35.03|

    DenseMixer consistently outperforms these baselines due to two key factors:

    (1) **Precision (vs. DefaultMoE):** It relies on historical approximations for inactive experts, while DenseMixer computes exact outputs.

    (2) **Compatibility (vs. ReMoE/SparseMixer):** These methods require model to be pretrained in the same way. Applying them *post-hoc* to standard TopK models creates structural misalignment, whereas DenseMixer is compatible with existing TopK mechanisms.

- **`Results on MiniCPM-8x2B`**

    As noted in our paper (L438-441), SparseMixer is designed for small K and specific pre-training regimes. To validate our implementation, we test with **MiniCPM-8x2B** (K=2) and SparseMixer showed gains in this low-$K$ setting.

    |Method|GSM|MBPP|HumanEval|Intent|Law|Sum.|Trans.|avg|
    |---|---|---|---|---|---|---|---|---|
    |Base|60.10|39.20|52.30|17.60|19.00|26.10|18.10|33.20|
    |Conv.|53.40|40.20|43.20|70.80|24.20|39.20|17.20|41.17|
    |DenseMixer|56.10|40.60|44.50|72.40|29.70|41.80|17.40|43.21|
    |SparseMixer*|42.20|34.68|40.82|63.40|23.30|29.70|16.80|35.08|



- **`Results on Deepseek v2 lite (16B)`**

    Due to time limit, we are only able to get the following results on **Deepseek v2 lite**.

    |Method|GSM|MBPP|HumanEval|Intent|Law|Sum.|Trans.|avg|
    |---|---|---|---|---|---|---|---|---|
    |Base|19.00|43.00|27.44|3.00|14.90|16.50|16.20|20.01|
    |Conv.|48.50|46.00|31.71|81.80|29.70|43.00|24.30|43.57|
    |DenseMixer|51.50|47.00|32.32|82.40|32.10|45.80|25.36|45.21|
    |DefaultMoE*|38.64|39.60|30.10|61.40|31.80|40.90|18.70|37.31|

---

**2. Hyperparameter Search Clarification**

The search ranges and selected optimal values (which fall inside the ranges) are detailed below:

|Model|Method|BS|LR|
|---|---|---|---|
|Qwen1.5-MoE|full|{32,64,128}|{5e-7,1e-6,2e-6,5e-6,1e-5,2e-5,3e-5,4e-5}|
| |lora|{32,64,128}|{5e-5,1e-4,2e-4,3e-4,4e-4,5e-4,6e-4,8e-4}|
|OLMoE|full|{64,128,256,512}|{5e-7,1e-6,2e-6,5e-6,1e-5,2e-5,3e-5,4e-5}|
| |lora|{128,256,512}|{5e-5,1e-4,2e-4,3e-4,4e-4,5e-4,6e-4,8e-4}|
|DeepseekV2-Lite|full|{32,64,128}|{5e-7,1e-6,2e-6,5e-6,1e-5,2e-5,3e-5,4e-5}|
| |lora|{32,64,128}|{5e-7,1e-6,2e-6,5e-6,1e-5,2e-5,3e-5,4e-5}|
|Qwen3-30B-A3B|full|{8,16,32}|{5e-6,1e-5,2e-5,3e-5,4e-5,5e-5}|

---

**3. Efficiency Considerations**

**Theoretical FLOPs**

While the FFN forward cost scales linearly with $E$, the **total** training overhead is sub-linear.
The total training FLOPs can be decomposed as:

$Total FLOPs=Forward_{Attn} + `Forward_{FFN}`+ Backward$

- In standard training, $Backward \approx 2 \times Forward$. DenseMixer only increases `$Forward_{FFN}$`. So the total training FLOPs does not grow linearly with expert number.
- Given smaller post-training datasets, the absolute overhead is marginal. For instance, on Qwen1.5-MoE, training time increased by only ~2 minutes (22-24 min), representing a highly favorable trade-off for the consistent performance gains.

**Wall-clock Time**

We use Qwen1.5-MoE with LLaMA-factory library and 4xH200 GPUs. As expected, DefaultMoE is slightly faster due to reduced FLOPs. However, for post-training, DenseMixer's absolute overhead is marginal—often just a few minutes—which we believe is a worthy trade-off for the superior performance.

|Dataset|Conv.|DenseMixer|DefaultMoE|
|---|---|---|---|
|Intent (7K)|22min|24min|22.7min|
|Law (1K)|8.5min|9.5min|8.9min|
|Summary (19K)|1.2h|1.4h|1.28h|
|Translation (11K)|39min|45min|42min|

---

**4. Summary**

Across all added experiments and analyses, DenseMixer demonstrates:

- **Superior accuracy** to DefaultMoE, ReMoE, and SparseMixer on models using mainstream Top-K routing,
- **Compatibility** with existing MoE checkpoints (no pretraining modifications),
- **Stable performance** under standard hyperparameter searches, and
- **Acceptable efficiency trade-offs** for the post-training setting.

We thank the reviewers again and hope the new results clarify DenseMixer’s empirical and practical advantages.

---

### Author Response · Authors · 2025-12-04
**Rebuttal Summary (after the Identity Leakage Issue)**

Our original scores were **2, 4, 6, 8**. During rebuttal, **Reviewer WUwb** (original score **4**) explicitly stated that all their concerns were fully addressed and **raised their score to “accept” (score of 6)** **`before the reviewer-identity leakage incident`**. So our scores should be **2, 6, 6, 8** before the rebuttal freezing and score resetting.

After the leak, ICLR globally rolled back all score changes, so the displayed score is again **4**, but the reviewer’s *written comment still recommends acceptance*.

During rebuttal, we focused on the key concerns and provided targeted new evidence:

- Added **direct comparisons** to **DefaultMoE, SparseMixer, and ReMoE** across Qwen1.5-MoE and DeepSeek-v2-Lite, with wall-clock and FLOP analysis.
- Confirmed that our **hyperparameters are interior points** in a non-trivial grid, and DenseMixer uses the **same tuned settings as baselines** (no extra tuning cost).
- Provided additional results showing **sub-linear overhead scaling** and small absolute time cost in post-training.
- Added **router-gradient statistics** demonstrating stronger and lower-variance gradients.
- Clarified the **non-triviality** of applying STE to modern Top-K and normalized Top-K MoEs.

After rebuttal (before the mechanical score rollback), the effective reviewer stance was **three reviewers recommending acceptance (≈6, 6, 8)** and **one reviewer (2) with novelty concerns**, while still acknowledging that the method is practical, sound, and empirically beneficial.

---

### Meta-Review · Area_Chair_UuZm · 2025-12-09

**Summary:**

The manuscript is reviewed by four reviewers, of whom one recommends rejection, two rated above the acceptance threshold (one of them after the rebuttal), and the other recommended acceptance.
The reviewers pointed out some weaknesses, such as lack of novelty and unclear experimental details. Moreover, one of the reviewers stated the paper's technical contribution is the application of the STE which is the key component of MoE routers. Two other reviewers suggested some additional experiments and ablation analysis that the authors addressed in the revised version of the paper.

After carefully considering the rebuttal and the revised paper. I agree with Reviewer 7J9M that the novelty of the work appears limited. Indeed, the idea of activating all experts during the forward pass and applying an STE to obtain router gradients is straightforward and related to existing MoE approaches that were studied in the paper.
Moreover, the paper does not sufficiently clarify what fundamentally differentiates DenseMixer from SOTA approaches or why this should be viewed as a new methodological contribution. Therefore, I believe the submission does not meet the standard required for acceptance.

**Reviewer Concerns:**

The authors address the issues raised by reviewers WUwb, 7J9M, VXEj, and 4vjf about the extra experiments and evaluations in comparison to SOTA approaches. However, 7J9M's concerns about the proposed model contributions are not satisfactorily addressed.

**Reviewer Scores:**

Reviewer WUwb acknowledged the willingness to raise the score. Reviewers 7J9M, VXEj and 4vjf will probably keep their current rating.

---

### Decision · Program_Chairs · 2026-01-26

Reject